# A Comparison of Different Intensive Care Unit Definitions Derived from the German Administrative Data Set: A Methodological, Real-World Data Analysis from 86 Helios Hospitals

**DOI:** 10.3390/jcm13123393

**Published:** 2024-06-10

**Authors:** Christina Bogdanov, Sven Hohenstein, Jörg Brederlau, Heinrich Volker Groesdonk, Andreas Bollmann, Ralf Kuhlen

**Affiliations:** 1University Hospital Leipzig, 04103 Leipzig, Germany; 2Helios Health Institute, 13125 Berlin, Germany; sven.hohenstein@helios-health-institute.com (S.H.); ralf.kuhlen@helios-health.com (R.K.); 3Helios Hospital Berlin Buch, 13125 Berlin, Germany; joerg.brederlau@helios-gesundheit.de; 4Helios Hospital Erfurt, 99089 Erfurt, Germany; heinrich.groesdonk@helios-gesundheit.de; 5Helios Hospital Leipzig, 04289 Leipzig, Germany; andreas.bollmann@helios-gesundheit.de

**Keywords:** ICU, ICU definition, ICU bed, ICU utilization, ICU admission, ICU capacity

## Abstract

**Background:** The intensive care unit (ICU) is a scarce resource in all health care systems, necessitating a well-defined utilization. Therefore, benchmarks are essential; and yet, they are limited due to heterogenous definitions of what an ICU is. This study analyzed the case distribution, patient characteristics, and hospital course and outcomes of 6,204,093 patients in the German Helios Hospital Group according to 10 derived ICU definitions. We aimed to set a baseline for the development of a nationwide, uniform ICU definition. **Methods**: We analyzed ten different ICU definitions: seven derived from the German administrative data set of claims data according to the German Hospital Remuneration Act, three definitions were taken from the Helios Hospital Group’s own bed classification. For each ICU definition, the size of the respective ICU population was analyzed. Due to similar patient characteristics for all ten definitions, we selected three indicator definitions to additionally test statistically against IQM. **Results**: We analyzed a total of 5,980,702 completed hospital cases, out of which 913,402 referred to an ICU criterion (14.7% of all cases). A key finding is the significant variability in ICU population size, depending on definitions. The most restrictive definition of only mechanical ventilation (DOV definition) resulted in 111,966 (1.9%) cases; mechanical ventilation plus typical intensive care procedure codes (IQM definition) resulted in 210,147 (3.5%) cases; defining each single bed individually as ICU or IMC (ICUᴧIMC definition) resulted in 411,681 (6.9%) cases; and defining any coded length of stay at ICU (LOSi definition) resulted in 721,293 (12.1%) cases. Further testing results for indicator definitions are reported. **Conclusions**: The size of the population, utilization rates, outcomes, and capacity assumptions clearly depend on the definition of ICU. Therefore, the underlying ICU definition should be stated when making any comparisons. From previous studies, we anticipated that 25–30% of all ICU patients should be mechanically ventilated, and therefore, we conclude that the ICUᴧIMC definition is the most plausible approximation. We suggest a mandatory application of a clearly defined ICU term for all hospitals nationwide for improved benchmarking and data analysis.

## 1. Introduction

Critical care plays a major role as a scarce resource in health care systems all around the world [1]. According to a statement by the World Federation of Societies of Intensive and Critical Care Medicine, the evolution of intensive care is driven by its capacity to provide rapid resuscitative and supportive care in critically ill patients [2]. For this purpose, it is crucial to have clarity on available capacities based on reliable, well-defined, and timely data [3]. International comparisons and global benchmarks for ICU performance are limited by differing ICU definitions resulting in a lack of generally comparable data. To date, there is no worldwide defined standard of critical care, nor is there a uniform definition of ICU [4,5,6]. In Germany, definitions of ICU are heterogeneous and the resulting data on ICU utilization vary significantly. The data basis to define ICU stays can range from the patient’s location in the hospital (i.e., length of stay at ICU), over legal principles and definitions of departments (§ 301 of the *Social Security Code Fifth Book Statutory Health Insurance* (*Sozialgesetzbuch Fünftes Buch Gesetzliche Krankenversicherung* (SGB V)), to ICU specific therapies and procedures (i.e., mechanical ventilation or specific operation and procedure codes like OPS 8-980, 8-98f) considering the hospital structure (i.e., number and qualification of the specialized staff to be provided and medical-technical equipment etc.) [7]. The scopes for these ICU definitions are rather different and hence, they bear the risk of misleading capacity assessments. Our study was driven by the COVID-19 pandemic response in Germany, with a strong national focus on securing ICU capacity to all acute patients in possible need for intensive care. Germany has one of the highest densities of ICU beds worldwide, with 35.3 ICU beds per 100,000 population capita [8]. ICU utilization is mainly dependent on the criteria for ICU admission and discharge, as well as the characteristics of the actual treatments provided in the respective ICU bed. Studies have found a high variability in ICU admission rates for a given acute condition among hospitals without evident impact on mortality, readmission rates, or costs [9]. In Germany, ICU utilization implies a mix of patients that can be distinguished into those undergoing typical ICU treatment (critically ill patients), such as mechanical ventilation or other acutely life-saving or organ-supporting therapies, and patients admitted to the ICU without undergoing specific intensive care treatment (patients with the need of expanded monitoring). Assumingly, those patients at risk are predominately monitored in the ICU. Intermediate care (IMC) or intermediate care units (IMCU) represent a “buffer” zone in hospitals either being a “step up” unit for patients exceeding the capacity of standard care or being a “step down” unit for patients who are too healthy to benefit from ICU treatments [10,11,12]. The German Hospital Statistics (2021) states that 57.5% of the German hospitals maintain ICU beds and a further 25.9% maintain IMC beds; 16.6% maintain neither [13]. Their structural organization within the hospital remains not well defined and a distinction between integrated or parallel ICU/IMC wards cannot be easily inferred from the data. This has consequences for adequate staffing especially regarding the regulatory required nursing capacity, potentially affecting patient safety and outcomes and it might cause further inefficiencies of resource utilization [11,14,15,16]. Finally, there is a lack of well-defined ICU admission criteria to allow a distinction from medical needs focusing on intensive care therapies and needs, like IMC, focusing on monitoring [11,17,18]. The goal of this study was to compare different ICU definitions by analyzing the impact of those definitions on the size of the resulting ICU population, patient characteristics, hospital course, and outcomes using administrative data from the 86 hospitals of the Helios Group over a period of 6 years. This study suggests the development of a clear definition for what is meant by an ICU that is mandatory for all hospitals nationwide.

## 2. Materials and Methods

We assessed ten different ICU definitions derived from the German administrative data set and compared the key parameters of the resulting population size and patient characteristics from 86 hospitals of the Helios Hospital Group, Germany. We derived three ICU definitions from the Helios hospitals, where they have an additional standardized and customized definition of ICU beds relying on the classification of each individual bed as an ICU or IMC bed. This ICU/IMC bed classification table is routinely filled by specialist staff and thus provides most timely, accurate and reliable overview of ICU beds within each hospital from 2016 to today. For the other seven definitions, we used claims data according to the German Hospital Remuneration Act (§21, Gesetz über die Entgelte für voll- und teilstationäre Krankenhausleistungen (KHEntgG)). The use of the Helios claims database is described in detail elsewhere [19]. Briefly, the data set contains admission and discharge data, age, sex, diagnoses, and comorbidities based on the International Classification of Disease, 10th Revision—German Modification (ICD-10-GM), medical and surgical procedures, treatments according to the Operation and Procedure Classification System (OPS), and a description of the wards where the patients have been treated during their hospital stay. In-hospital mortality, length of stay in the hospital (LOSh), as well as in an intensive and/or intermediate care unit (ICUᴧIMC—in one unit; ICU; IMC) were calculated, respectively [20]. We analyzed inpatient cases of patients ≥ 18 years of age being admitted to ICU in the time period from 1 January 2016 until 31 December 2021. We differentiated between all patients in the dataset (all cases) and “completed cases”. Completed cases excluded patients who were transferred to another hospital. In-hospital mortality rate was defined as the number of deaths as the reason for hospital discharge divided by all completed cases. LOSh was defined as the day of admission plus every other day of hospitalization, excluding the day of transfer or discharge. The time of attendance is documented at midnight each day of the patient’s stay and is accompanied by the departmental designation. When a patient is admitted, transferred, or discharged on the same day, this day is considered the admission day [21]. LOSh was defined as documented occupancy stay >1 day within all completed cases.

We used the ten ICU definitions described in Table 1 including the respective reason for choice per each definition. Definitions 1–7 were derivable from the routine data set, definitions 8–10 were taken from the Helios hospital bed classification.

For each ICU definition, we analyzed the size of the ICU population as the number of completed cases meeting the respective criteria. Based on the absolute population and their percentage of the total hospital population, we will select indicator definitions to test statistically against IQM because it is validated in numerous publications. The statistical analysis of the different case numbers per indicator definition was performed by estimating the incidence–rate ratio (IRR), with ICU admission according to the respective definition being the incident. We further compared the patient characteristics for the different indicator definitions with respect to age, sex, Elixhauser comorbidity index, treatment episodes with extracorporal membrane oxygenation (ECMO; 8-852.0/8-852.3/8-852.6), in-hospital mortality rate, and length of stay (LOSh). Inferential statistics were based on generalized linear mixed models (GLMM) specifying hospitals as a random factor [22]. We employed a logistic GLMMs function for dichotomous data, Poisson GLMMs for count data, and LMMs for continuous data. Effects were estimated with the lme4 package (version 1.1-26) in the R environment for statistical computing (version 4.0.2, 64-bit build) [23,24]. For all tests we apply a two-tailed 5% error criterion for significance. For the description of the patient characteristics of the cohorts, we employed χ2-tests for binary variables and analysis of variance for numeric variables. We report proportions, means, standard deviations, and *p* values. For the comparison of selected treatments and outcomes in the different cohorts, we used logistic GLMMs with a logit link function. We report proportions and odds ratios together with confidence intervals and *p* values. The analysis of the outcome variable length of stay (LOSh) was performed via LMMs. For these analyses, we log-transformed the dependent variable due to its skewed distribution. We report means, standard deviations, and *p* values. The computation of *p* values is based on the Satterthwaite approximation for degrees of freedom. For the weighted Elixhauser comorbidity index, the AHRQ algorithm was applied [25,26]. 

## 3. Results

A total of 6,204,093 patient treatment episodes were analyzed during the 6-year study period. Of these, 5,980,702 cases completed the hospital stay and 913,402 cases fulfilled at least one of the chosen ICU criteria. Case development over the years is depicted in Table 2. 

For each ICU definition, we analyzed the size of the ICU population as the number of completed cases meeting the respective criteria. The case number for all definitions, patient characteristic, treatments, and clinical outcomes are depicted in Table 3.

The main finding of our study is a significant variability of ICU population size ranging from 111,966 (representing 1.9% of all completed cases) to 730,293 (12.2%) completed cases depending on definition. Due to similar results in patient characteristics, we identified three clusters based on the absolute population and their percentage of the total hospital population: cluster 1 included the definitions with the largest populations LOSivDOV (730,293/12.2%), LOSivOPS (725,394/12.1%), and LOSi (721,293/12.1%); cluster 2 included the definitions with the smallest populations OPS (210,259/3.5%), IMC (125,259/2.5%), and DOV (111,966/1.9%); cluster 3 included the mid-range populations of ICUᴧIMC (411,681/6.9%), ICU (237,647/4.0%), and Department (234,926/3.9%). The definition IQM (238,500/4.0%) was considered separately because it is validated in numerous publications. For statistical analysis, we selected indicator definitions per cluster: In cluster 1, LOSi was chosen because it represents the depiction of patients with an ICU need without including the control variables DOV and OPS; in cluster 2, DOV was chosen because it is the hardest factor for ICU need; in cluster 3, ICUᴧIMC was chosen because the population of this definition is composed of real and robust care data from the Helios hospitals. Elixhauser comorbidity index, IRR, ECMO treatment, and in-hospital mortality rate were significantly different for the indicator definitions. In contrast, age, sex, and length of stay (LOSh) are rather independent from the ICU definitions (Table 4). The comparison of the Elixhauser comorbidity index distributions within each definition shows the largest proportion of patients in the category ≥ 5 in every indicator definition ranging from 88.7% (99,369) in DOV to 67.8% (310,499) in ICUᴧIMC, all with a *p*-value < 0.01. The comparison of daily admission rates shows a range from DOV with an IRR of 0.59 (0.55–0.64) to ICUᴧIMC with an IRR of 2.09 (1.86–2.33), all with a *p*-value < 0.01. The comparison among the indicator definitions showed LOSi 69.3% (499,558) and ICUᴧIMC 67.8% (310,499) have on average about 17% less patients in the ≥5 category, while having significantly more patients in the category < 0 (ICUᴧIMC (15.6% (71,383); LOSi (14.3% (102,951)). The highest in-hospital mortality rate compared to IQM with 23.5% (55,936)) was in DOV (odds ratio 2.29 (2.25–2.32)), followed by LOSi (odds ratio 0.35 (0.34–0.35)), and ICUᴧIMC (odds ratio 0.32 (0.32–0.33), all with *p*-value < 0.001). The highest likelihood of being treated with ECMO compared with IQM (ECMO: 0.8% (1819)) is in DOV (odds ratio of ECMO: 2.42 (2.26–2.59)), followed by LOSi (odds ratio of ECMO: 0.36 (0.34–0.39)) and ICUᴧIMC (odds ratio of ECMO: 0.29 (0.25–0.32; all with *p*-value < 0.001).

## 4. Discussion

The goal of this study was to compare different ICU definitions and the impact on resulting ICU population size, patient characteristics, hospital course, and clinical outcomes. We found that different ICU definitions lead to significantly different ICU population sizes ranging from 1.8% (111,966 cases) to 12.2% (730,293 cases) of all completed hospital cases. This denominator variability limits the comparability of results for any relative measure for outcomes or procedures and therapies, thereby influencing all comparisons of numbers like these. Capacities and their utilization might thereby be either significantly overestimated or underestimated. The COVID-19 pandemic has brought national governance of intensive care capacity into focus [27,28]. Numerous study results state that 25–30% of intensive care patients are being mechanically ventilated [29,30,31]. These were represented in the most rigid DOV definition with 111,966 cases, where only cases being on a ventilator were counted. Extrapolating ventilated patients to an ICU share of 25–30%, around 300,000–350,000 cases should be expected as valid number for all ICU cases. The closest definition to this extrapolation was ICUᴧIMC, with 411,681 cases. In comparison IQM, only reflecting different ICU-related OPS codes and/or duration of ventilation > 0 h, counts 238,500 cases, which implies an underestimation. This largely variable, subjective understanding of an ICU is coherent with previous studies [32,33]. Marshal et al. (2017) analyzed the international intensive care landscape and concluded that there was no clear categorization, nor a set of worldwide valid descriptive criteria defining an ICU [2]. The better the characteristics of an ICU are reflected within the data set, the more structured, accurate, and quickly retrievable the databases, with available capacity, can be designed [34]. Wunsch et al. (2008) studied the variation in critical care services across North America and Western Europe and state the need for accurate critical care data for reliable interpretations [35]. Ensuring actual comparability and transparency via an improved data basis might lead to a more robust assessment of the performance and quality of German hospitals [2,36,37].

The heterogeneity of ICU definitions bears the risk of unbeneficial ICU bed utilization. Studies have shown that neither the use of a surgical procedure itself nor a medical diagnosis per se accurately define the need for ICU, measured as clinical outcome benefit [38]. Instead, a more usable definition seems to be based on the acute condition of a patient, resulting from a reversible pathology and a proven benefit of ICU interventions [39]. Truog (1992) indicates a threshold health status of “too good” and “too bad” to benefit from intensive medical treatment, both with a low ‘benefit index’ due to their either very low or very high likelihood to recover with or without intensive care [40]. Studies show that if the benefit index is not considered, an allocation risk exists by unnecessarily blocking beds for those patients who would benefit from intensive care treatment [17,18,41]. Further, Stang et al. (2020) stated a non-uniform occupancy of ICU beds due to no nationwide coordination of ICU capacities, which leads to some hospitals being overloaded while others remain rather empty [42]. Hence, the question of to what extent ICU admissions were based on general diagnoses or procedures rather than actual medical necessity needs to be answered instead of merely assessing capacity, which will always be too little when not used appropriately.

A uniform definition, when elaborated, can be transferred, and applied directly to the DRG browser: All hospital individual §21 data sets, including hospital structural data, as well as performance data, are pseudonymized and delivered to the central Diagnosis Related Group (DRG) data center where the data is checked for completeness and formal plausibility. The billing data is calculated in accordance with the methodology for calculating treatment costs—the *Calculation Manual of the Institute Version 4.0* of the Hospital Remuneration System (*Institut für das Entgeltsystem im Krankenhaus* “InEK”) [43]. The data set is publicly accessible in the DRG browser and is used for the further development of the DRG system to improve health system performance as institutional quality management [44,45]. 

Our study has several methodical limitations: 

(1) The selected definitions are not clearly replicable nor controllable. The list of approaches is not necessarily finite. For lack of an ICU definition, IQM was adopted as the first quality definition and the other nine were defined as surrogates that could be derived from routine data in regard with the Hospital Remuneration Act, Section 6, Paragraph 21 (Gesetz über die Entgelte für voll- und teilstationäre Krankenhausleistungen (KHEntgG)) and taken from the Helios Hospital Group’s own bed classification. According to this, hospitals are required to provide information on their completed services; here, challenges lie in the same understanding of what is meant by ICU—disparities are to be expected [46]. Thus, the selected ten definitions represent a range of possible definitions tested by comparison with studies from other countries. Certainly, there are other ways to define ICU. 

(2) A delta is to be expected between the services that were provided to the patient and those services that can be billed and can therefore be included in the analysis within the definition Department. Yet, it is a relatively robust source of data on performed ICU treatments, it creates transparency due to the transmission of performance data, and it is relevant to hospital billing [47].

(3) Although the IMC definition is not an ICU definition in essence, we have included it in our consideration for two reasons: (a) as a buffer zone between the normal ward and the ICU, it is an important structural unit that is dependent on intensive care, and (b) many ICUs are not sharply separated from IMC. The unique characteristic of the Helios example is the clearly indicated flexible use of beds, so its inclusion seems justified. We have tested the individual consideration of this definition based on an eleventh test definition defined with cases from IMC, ICU, and ICUᴧIMC. The sum of these cases exceeded 800,000, so multiple counting of the same cases was proven. 

(4) We selected one representative definition from each cluster for statistical testing against IQM. We consider our selection to be a resiliently plausible one. Another selection would also be possible.

(5) Despite the existence of many charge catalogs and billing regulations, our results indicate a systematic coding error. In this study, we considered several legal bases and regulations regarding coding as Krankenhausentgelt Gesetz (KHEntgG), KFPV, and Sozial Gesetzbuch (SGB) V §301. In addition, there are the German Coding Guidelines, which set out certain rules for coding diagnoses and procedures, as well as guidance on the application of procedure classifications, e.g., the ICU definition. LOSi is part of the documentation requirements for recording intensive medical complex treatment by the German Institute for Medical Documentation and Information (DIMDI) as part of the German Federal Institute for Drugs and Medical Devices (BfArM) [48]. Thus, it seems that LOSi is not used at all or is not plausibly seen by the imbalanced coding of LOSi compared to DOV. This leads to a reduction in data quality. Given the level of detail of the legislation and the observations in this study, the question of correct implementation arises. 

(6) The introduction of a uniform definition of ICU with clear criteria will not replace or override the clinical assessment and individual needs of a patient.

(7) The introduction of a uniform, national ICU definition is the responsibility of the legislator. Initial steps can be taken at an individual hospital level, but only a legal requirement can create a common definition basis.

## 5. Conclusions

Different ICU definitions result in different population sizes and generate various denominators, such that all outcomes and relative data are difficult to compare. Comparable results can be obtained if robust ICU markers are used. From previous studies, we anticipated that 25–30% of all ICU patients should be mechanically ventilated, and therefore, we conclude that the ICUᴧIMC definition is the most plausible approximation. This study suggests the development of a clear definition for what is meant by an ICU that is mandatory for all hospitals nationwide. This offers the added value of a time-independent, reliable, and immediate query of nationwide ICU capacities—within seconds, as an improved (inter-)national benchmark.

## Figures and Tables

**Table 1 jcm-13-03393-t001:** Ten derived ICU definitions.

Nr	Abbreviation	Definition and Its Criteria	Reason for Choice	Total Number of Completed Cases per Definition (n) and Proportion of All Completed Cases (%)
1	IQM	Initiative for Quality in Medicine ⋅ OPS 8-980, 8-98f and/or ⋅ Duration of ventilation > 0 h	⋅ reference definition ⋅ is used in numerous publications ⋅ contains two robust indicators for intensive care treatment ⋅ relevant to hospital revenue	238,500 (4.0%)
2	DOV	Duration of mechanical ventilation ⋅ Duration of mechanical ventilation > 0 h	⋅ hard indicator for intensive care treatment at an ICU ⋅ relevant to hospital revenue	111,966 (1.9%)
3	OPS	Intensive care Operation and Procedure Code ⋅ Intensive care Operation and Procedure Code 8-980 and/or ⋅ Intensive care Operation and Procedure Code 8-98f	⋅ official classification for coding operations, procedures, and general medical measures ⋅ a robust indicator of performed intensive care treatment ⋅ relevant to hospital revenue	210,147 (3.5%)
4	Department	Department ⋅ § 301 of the Social Security Code Fifth Book—Statutory Health Insurance (SGB V) for accounting purposes	⋅ legally regulated data transmissions from hospitals to health insurances ⋅ a relatively robust source for data on intensive care treatments ⋅ transmission and preparation of performance data, creates data transparency ⋅ relevant to hospital billing	234,926 (3.9%)
5	LOS_i_	Length of Stay at ICU Length of Stay > 0 days at ICU	⋅ indicate a high need for care ⋅ display patients’ location at ICU ⋅ are part of the documentation requirements for recording intensive medical complex treatment by the German Institute for Medical Documentation and Information as part of the German Federal Institute for Drugs and Medical Devices	721,293 (12.1%)
6	LOS_i_vOPS	Length of Stay at ICU or coding of Intensive Care Operation and Procedure Code ⋅ Length of Stay > 0 days at ICU or Intensive care Operation and Procedure Code 8-980 and/or 8-98f		725,394 (12.1%)
7	LOS_i_vDOV	Length of Stay at ICU or duration of mechanical ventilation ⋅ Length of Stay > 0 days at ICU or ⋅ Duration of mechanical ventilation > 0 h		730,293 (12.2%)
8	ICU	Helios hospital Group’s own bed classification: ⋅ Classified hospital bed as ICU bed	⋅ indicate intensive care treatment ⋅ provides an overview of each ICU designation and the nursing location	237,647 (4.0%)
9	IMC	Hospital Section Intermediate Care Unit ⋅ Classified hospital bed as IMC bed and ⋅ Not classified hospital bed as ICU bed ⋅ Not classified hospital bed as ICU/IMC (combined) bed		152,259 (2.5%)
10	ICUᴧIMC	Hospital section intensive care unit and intermediate care unit combined ⋅ Classified hospital bed as ICU bed and ⋅ Classified hospital bed as IMC bed		411,681 (6.9%)

**Table 2 jcm-13-03393-t002:** Development over the years of inpatient cases.

Cases	All Inpatient Cases	All Completed Cases	All Cases Referring to Any ICU Criterion
2016	1,068,610	1,030,388	172,140
2017	1,076,906	1,036,797	165,646
2018	1,071,445	1,030,591	153,076
2019	1,073,693	1,037,903	147,159
2020	963,883	929,960	141,095
2021	949,556	915,063	134,286
Sum of 6 years	6,204,093	5,980,702	913,402

**Table 3 jcm-13-03393-t003:** Overview of case count and characteristics per definition.

Definition	Number of Completed Cases (Proportion and Average Yearly IC Numbers)	Age (Mean)	Sex (Proportion Female)	In-Hospital Mortality	ECMO	Length of Stay (LOSh Mean)
LOSivDOV	730,293 (12.2%) 121,716 ± 12,225	69.2 ± 15.3	43.8% (319.,868)	10.3% (75,220)	0.26% (1899)	11.3 ± 12
LOSivOPS	725,394 (12.1%) 120,899 ± 12,480	69.2 ± 15.3	43.8% (317,723)	9.5% (68,912)	0.24% (1741)	11.4 ± 12
LOSi	721,293 (12.1%) 120,216 ± 12,024	69.2 ± 15.3	43.8% (315,962)	9.4% (67,802)	0.24% (1731)	11.4 ± 12
ICUᴧIMC	411.681 (6.9%) 68,614 ± 12,842	69.1 ± 15.9	45.0% (185,256)	9.0% (37,051)	0.09% (371)	10.8 ± 11.2
IQM	238,500 (4.0%) 39,750 ± 7090	69.6 ± 14.2	41.1% (98,024)	23.5% (56,047)	0.76% (1813)	15.6 ± 16.1
ICU	237,647 (4.0%) 39,608 ± 5012	68.2 ± 15.2	41.6% (98,861)	16.1% (38,261)	0.65% (1545)	12.3 ± 13.6
Department	234.926 (3.9%) 39,154 ± 5759	67.5 ± 15.3	40.3% (94,675)	12.1% (28,426)	0.61% (1433)	12.7 ± 13.9
OPS	210,147 (3.5%) 35,024 ± 7066	69.3 ± 14.2	40.8% (85,740)	19.9% (41,819)	0.77% (1618)	16.4 ± 16.4
IMC	152,259 (2.5%) 25,376 ± 1268	69.3 ± 15.4	45.9% (69,887)	3.2% (4872)	0.01% (15)	8.5 ± 9.1
DOV	111,966 (1.9%) 18,661 ± 578	70.1 ± 13.4	40.5% (45,346)	41.4% (46,354)	1.58% (1769)	17.4 ± 19

**Table 4 jcm-13-03393-t004:** Indicator definitions’ cohort comparison.

**Patient Characteristics**					
**Cohort**	**IQM**	**LOSi**	**DOV**	**ICUᴧIMC**	***p* Value**
**Age**					
Mean (SD)	69.6 ± 14.2	69.2 ± 15.3	70.1 ± 13.4	68.9 ± 16.0	<0.01
18–59 years	21.8% (52,058)	23.8% (171,635)	19.8% (22,141)	24.6% (112,574)	<0.01
60–69 years	22.3% (53,085)	20.6% (148,586)	23.6% (26,417)	19.6% (89,754)	<0.01
70–79 years	28.9% (68,929)	27.1% (195,456)	29.8% (33,391)	26.4% (120,683)	<0.01
≥80 years	27.0% (64,428)	28.5% (205,616)	26.8% (30,017)	29.4% (134,706)	< 0.01
**Sex**					
Male	58.9% (140,480)	56.2% (405,074)	59.5% (66,640)	55.0% (251,826)	
Female	41.1% (98,020)	43.8% (316,219)	40.5% (45,326)	45.0% (205,891)	<0.01
**Elixhauser comorbidity index**					***p* value**
Mean (SD)	17.2 ± 12.8	12.4 ± 12.2	19.8 ± 12.6	12.1 ± 12.3	<0.01
<0	7.1% (17,016)	14.3% (102,951)	3.8% (4225)	15.6% (71,383)	<0.01
0	4.3% (10,250)	7.9% (57,299)	3.0% (3324)	9.0% (41,279)	<0.01
1–4	5.6% (13,347)	8.5% (61,485)	4.5% (5048)	7.5% (34,556)	<0.01
≥5	83.0% (197,887)	69.3% (499,558)	88.7% (99,369)	67.8% (310,499)	<0.01
**Admission rates**					
**IRR**	**Total**	**Daily admissions**	**IRR (95% CI)**	***p* value**	
IQM	238,500	108.8	-	-	
LOSi	721,293	329.1	5.97 (5.55–6.42)	<0.01	
DOV	111,966	51.1	0.59 (0.55–0.64)	<0.01	
ICUᴧIMC	411,681	187.8	2.09 (1.86–2.33)	<0.01	
**Treatments and Outcomes**					
**Cohort**	**Proportion (*n*)**	**Odds Ratio** **(95% CI)**	***p* value**		
**Extracorporeal membrane oxygenation (ECMO)**					
IQM	0.8% (1819)	-	-		
LOSi	0.2% (1766)	0.36 (0.34–0.39)	<0.001		
DOV	1.6% (1768)	2.42 (2.26–2.59)	<0.001		
ICUᴧIMC	0.1% (365)	0.29 (0.25–0.32)	<0.001		
**In-hospital mortality**					
IQM	23.5% (55,936)	-	-		
LOSi	9.4% (68,114)	0.35 (0.34–0.35)	<0.001		
DOV	41.4% (46,385)	2.29 (2.25–2.32)	<0.001		
ICUᴧIMC	9.0% (36,878)	0.32 (0.32–0.33)	<0.001		
**Length of stay (LOSh)**					
**LOSh**	**Mean (SD)**	**Coefficient** **(95% CI)**	***p* value**		
IQM	15.6 ± 16.1	-	-		
LOSi	11.4 ± 12.0	−0.21 (−0.21–−0.21)	<0.01		
DOV	17.4 ± 19.0	0.00 (−0.01–0.00)	0.62		
ICUᴧIMC	10.8 ± 11.2	−0.23 (−0.24–−0.23)	<0.01		

## Data Availability

The data presented in this study are available upon request from the corresponding author due to legal reasons.

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
