# Peer review of "A Comparison of Different Intensive Care Unit Definitions Derived from the German Administrative Data Set: A Methodological, Real-World Data Analysis from 86 Helios Hospitals"

_jcm, 2024, doi:10.3390/jcm13123393_

Round 1

Reviewer 1 Report

Comments and Suggestions for Authors

Authors investigated highly important problem of defining ICU stay and occupacy in hospitals. The analysis of different definitions and approaches to it.

The article is of high importance and emphasize necessarity of ICU - IMC distinction as well as importance to have step down path to ward care in order to unburden ICU.

I have one question that needs explanation:

In Introduction, line 75 authors stated:... The German Hospital Statistics 75 (2021) states that 57.5% of the German hospitals maintain ICU beds and further 25.9% IMC beds [13].

Does it mean that 25.9% of hospitals has IMC beds?

And 57.5% of hospital has ICU beds. If understanding correctly, there are hospitals without ICU beds?. Please explain

Reviewer 2 Report

Comments and Suggestions for Authors

Thank you for the opportunity to review this paper.  A very elaborate study trying to come up with the criteria for the ICU stay and has reviewed a significant number of patient charts for the study. From my view placing the patient in the ICU vs IMC vs floors is more of the clinical perspective rather than the criteria. There are few gold standard criteria which makes the patients should go to the ICU like mechanical ventilation, requiring pressors, inotropes but most of them is more of the clinical judgement, nursing standards of the care for that particular hospital rather than the criteria by itself. Sometimes depending on the hospital standards, some patients can be intermediate care, but the same patient can be an ICU patient at the different hospital. The paper has the below defects. 1. It's a negative study as it doesn’t have new conclusive results than we already know 2. It's difficult to do the study to create an overall generalized criteria for the all the patients in the hospital for the ICU admissions as it is more clinical evaluation based, nursing standards of care for the hospital in particular and comfort level of the floor physicians to treat particular patient in the ICU vs floors. Also in the post op patients it varies as the same patient with the same procedure can be sent to the floors if they didn’t had any complications during the procedure vs had some complications requiring closer monitoring by the ICU level of care.

Reviewer 3 Report

Comments and Suggestions for Authors

The paper deals with the complexity of definition of ICU within the German health system, based on a survey conducted in the HELIOS hospital group. The point in determining this definition is to better evaluate medical procedures and costs within the system. 

Overall, the issue is well presented by the authors and highlights the problem in the classification of ICU structures. However, a central regulation that would clearly identify a structure as an ICU should take place  by the German health policy officials, and this would  more definitely solve the issue. For this reason, I find the paper of average clinical ineterest. 

Comments on the Quality of English Language

Minor English editing required. 
